
# Positive Matrix Factorization of Organic Aerosol: Insights from a Chemical Transport Model

Anthoula D. Drosatou[1,2], Ksakousti Skyllakou[2], Georgia N. Theodoritsi[1,2] and Spyros N. Pandis[1,2,3]

[1]Department of Chemical Engineering, University of Patras, Patras, Greece

[2]Institute of Chemical Engineering Sciences, Foundation for Research and Technology Hellas (FORTH/ICE-HT), Patras, Greece

[3]Department of Chemical Engineering, Carnegie Mellon University, Pittsburgh, PA 15213, USA

*Correspondence to*: Spyros N. Pandis (spyros@chemeng.upatras.gr)

## Abstract

Factor analysis of Aerosol Mass Spectrometer measurements (organic aerosol mass spectra) is often used to determine the sources of organic aerosol (OA). In this study we

aim to gain insights regarding the ability of positive matrix factorization (PMF) to identify and quantify the OA sources accurately. We performed PMF and multilinear engine (ME-2) analysis on the predictions of a state-of-the-art chemical transport model (PMCAMx-SR) during a photochemically active period for specific sites in Europe in an effort to interpret the diverse factors usually identified by PMF analysis of field

measurements. Our analysis used the predicted concentrations of 27 OA components, assuming that each of them is "chemically different" from the others.

The PMF results based on the chemical transport model predictions are quite consistent (same number of factors and source types) with those of the analysis of AMS measurements. The estimated uncertainty of the contribution of fresh biomass burning is

less than 30% and of the other primary sources less than 40%, when these sources contribute more than 20% to the total OA. For contributions between 10 and 20% the corresponding uncertainties increase to 50%. Finally, when these sources are small (less than 10% of the OA) the corresponding error is a factor of two or even three.

One of the major questions in PMF analysis of AMS measurements concerns the

sources of the two or more oxygenated OA (OOA) factors often reported in field studies.



Our analysis suggests that these factors include secondary OA compounds from a variety of anthropogenic and biogenic sources and do not correspond to specific sources. Their characterization in the literature as low and high volatility factors is probably misleading, because they have overlapping volatility distributions. However, the average volatility of

the one often characterized as low-volatility factor is indeed lower than that of the other (high volatility factor). Based on the analysis of the PMCAMx-SR predictions, the first oxygenated OA factor includes mainly highly-aged OA transported from outside Europe, but also highly aged secondary OA from precursors emitted in Europe. The second oxygenated OA factor contains fresher SOA from volatile, semi-volatile, and

intermediate volatility anthropogenic and biogenic organic compounds. The exact contribution of these OA components to each OA factor depends on the site and the prevailing meteorology during the analysis period.

## 1. Introduction

Exposure to high levels of fine atmospheric particles results in increased mortality and morbidity (Pope et al., 2009). The same particles affect climate by scattering and absorbing solar radiation (Seinfeld and Pandis, 2006), and also influence the properties and lifetime of clouds (IPCC, 2014). Organic aerosol represents an important fraction (20 to 90%) of fine particulate matter (Kanakidou et al., 2005; Zhang et al., 2007) and is

generated from biogenic and anthropogenic sources (de Gouw and Jimenez, 2009). It is usually characterized as primary (POA) when it is emitted directly in the particulate phase and secondary (SOA) when formed during the atmospheric oxidation of volatile, intermediate volatility, and semivolatile organic components.

The aerosol mass spectrometer (AMS) is a state-of-the-art instrument that can

measure continuously the fine OA concentration providing at the same time unit or high resolution mass spectra of the OA. These spectra can be used in factor analysis to acquire information about OA sources, processes, and properties (Zhang et al., 2011). Several factor analysis techniques have been developed to estimate the contributions of sources and processes to the observed OA. These techniques include custom principal component

analysis (Zhang et al., 2005), multiple component analysis (Zhang et al., 2007), positive



matrix factorization (PMF) (Paatero and Tapper, 1994; Lanz et al., 2007) and the multilinear engine (ME-2) (Paatero, 1999; Lanz et al., 2008; Canonaco et al., 2013).

Zhang et al. (2005) separated the OA in Pittsburgh into an oxygenated OA factor (OOA) associated with secondary sources and a hydrocarbon-like OA factor (HOA) that

represents POA related with urban sources and fossil fuel combustion. Lanz et al. (2007) identified additional important primary sources like biomass burning OA (bbOA). Measurements in Beijing showed that coal combustion (CCOA) is a major primary source in that area (Sun et al., 2013). Allan et al. (2010) identified cooking OA (COA) as a significant component of urban OA. However, Dall'Osto et al. (2015) argued that the

interpretation of the COA factor may be problematic as it may include OA from other sources and not just cooking. Kostenidou et al. (2018) also argued that the bbOA factor determined in the South US by Xu et al. (2017) may include oxygenated OA from other sources. Yuan et al. (2012) suggested that PMF factors may correspond to different stages of photochemical processing, rather than to independent sources. Aiken et al.

(2009) found that PMF can also yield factors that represent more than one source, especially in heavily polluted areas, due to their complex emission patterns. Brinkman et al. (2006) reported that when contributions from a pair of sources, such as diesel and gasoline exhaust, were highly correlated in synthetic datasets, a single factor corresponding to both sources was usually found. Despite these advances the accuracy of

the PMF-determined primary organic sources remains an issue of debate.

OOA represents a significant fraction of OA at many locations (Zhang et al., 2007). Lanz et al. (2007) further separated OOA into more oxygenated OA (OOA-1) and less oxygenated OA (OOA-2) during summer in Zurich. Ulbrich et al. (2009) also reported an OOA-1 and an OOA-2 factor in Pittsburgh repeating the original analysis of

Zhang et al. (2007). Typically, PMF of ambient AMS data identifies two types of OOA: a more oxidized OOA factor (OOA-1) which is thought to be more aged and almost non-volatile and a less oxidized factor (OOA-2) which is thought to be semivolatile (Jimenez et al., 2009; Ng et al. 2010). Huffman et al. (2009a) have showed that OOA-2 is usually more volatile than OOA-1 and includes less oxygenated secondary material (Jimenez et

al., 2009). Jimenez et al. (2009) used the acronyms LV-OOA (low volatility) and SV-OOA (semivolatile) for OOA-1 and OOA-2, respectively. Paciga et al. (2016) using





volatility measurements in Paris confirmed that SV-OOA is more volatile on average than LV-OOA, but argued that they both contain components with a wide range of overlapping volatilities. Kostenidou et al. (2015) proposed that the use of the SV-OOA

and LV-OOA may be misleading and used the terms very oxygenated OA (V-OOA) and moderately oxygenated OA (M-OOA). Hildebrandt et al. (2010) based on measurements in Finokalia, Greece proposed that the two OOA factors represent the more and less oxidized states of secondary OA during the period of the analyzed field measurements. They suggested that in remote areas during photochemically active periods the two OOA

factors are quite similar to each other as the OA is always at a very aged state. Other interpretations of the two OOA factors have also been proposed. For example, the less oxidized OOA (OOA-2) appeared to resemble biogenic SOA (bSOA) and the more oxidized OOA (OOA-1) appeared to be associated with transported OA from other areas in a study in Canada (Kiender-Schar et al., 2009; Sun et al., 2009).

Modeling efforts have so far focused on the comparisons of the factor analysis results of AMS measurements and the concentrations of modeled OA (Hodzic et al., 2010; Fountoukis et al., 2014; Tsimpidi et al., 2016). All these studies implicitly assume that each factor determined by PMF analysis of the AMS measurements corresponds to one group of sources.

In this work, we apply PMF analysis to the OA predictions of a chemical transport model in order to investigate whether PMF is able to separate the OA components from different sources or processes. Our main objective is to gain insights about the nature of the primary (POA, bbOA, etc.) and secondary (OOA-1, OOA-2, etc.) factors often determined in field studies and to quantify the corresponding uncertainties.

Our analysis assumes that each OA component in the model is chemically different than the rest. This is not the case in reality as different OA components may have similar AMS spectra. As a result, our analysis represents to some extent a best case scenario. However, the fact that the true sources and processes are known in this case makes this approach potentially useful.




## 2. Model Description

### 2.1 PMCAMx-SR

The model used in this study is the three-dimensional regional CTM PMCAMx-SR (Theodoritsi and Pandis, 2018), a regional three-dimensional CTM. PMCAMx-SR was applied to a $5400 \times 5832$ km$^2$ region covering Europe with $36 \times 36$ km grid resolution and 14 vertical layers extending up to 6 km. The model was set to perform simulations on a rotated polar stereographic map projection. The necessary inputs to the model include

horizontal wind components, temperature, pressure, water vapor, vertical diffusivity, clouds, and rainfall. All meteorological inputs were created using the meteorological model WRF (Weather Research and Forecasting) (Skamarock et al., 2005).

The gas-phase chemical mechanism of PMCAMx-SR is based on an updated version of the SAPRC mechanism with 211 reactions of 56 gases and 18 radicals

consisting of the gas-phase oxidation of semivolatile organic compounds (SVOCs), intermediate volatility organic compounds (IVOCs) and volatile organic compounds (VOCs). The OA composition is described in PMCAMx-SR using a set of lumped species distributed across a VBS (Donahue et al., 2006) with volatility bins (surrogate species) that have saturation concentration $C^*$ ranging from 0.01 to $10^6$ μg m$^{-3}$ separated

by one order of magnitude at 298 K. Primary organic compounds are all considered to be semi-volatile with $C^*$ ranging from $10^{-2}$ to $10^6$ μg m$^{-3}$ at 298 K (Shrivastava et al., 2008). Anthropogenic SOA (aSOA) and biogenic SOA (bSOA) are described separately using four volatility bins (1, 10, 100, 1000 μg m$^{-3}$). The secondary organic aerosol (SOA) formation and growth follows Murphy and Pandis (2009). The SOA module

incorporates NO$_x$-dependent SOA yields (Lane et, al. 2008b) and contains anthropogenic aerosol yields based on the studies of Ng et al. (2006) and Hildebrandt et al. (2009). The volatility distribution proposed by Shrivastava et al. (2008) was used assuming that the mass of IVOC emissions is approximately equal to 1.5 times the primary organic aerosol emissions (Robinson et al., 2007; Tsimpidi et al., 2010). This POA volatility distribution

is used in PMCAMx-SR for all sources with the exception of biomass burning. PMCAMx-SR simulates separately the fresh biomass burning organic aerosol (bbPOA) and its secondary oxidation products (bbSOA) using the volatility distribution of May et al. (2013) for the corresponding emissions.



Chemical aging in PMCAMx-SR is simulated assuming that the dominant
pathway is gas-phase oxidation of the corresponding organic compounds by OH,
assuming a rate constant equal to $1 \times 10^{-11}$ cm$^3$ molec$^{-1}$ s$^{-1}$ for anthropogenic SOA
components and $4 \times 10^{-11}$ cm$^3$molec$^{-1}$ s$^{-1}$ for the primary OA components and IVOCs
(Murphy and Pandis, 2009). Each reaction leads to one order of magnitude reduction of
the volatility of compound. The increase of the OA concentration due to the chemical
aging of biogenic SOA (bSOA) is assumed to be negligible.

The simulation period is May 2008, a warm summer-like month for most of
Europe. This period was selected because PMCAMx has been evaluated against
measurements from the EUCAARI campaign that took place during that month
(Fountoukis et al., 2011). Fountoukis et al. (2014) in a subsequent study found
encouraging agreement between predictions of PMCAMx and ME-2 analysis of AMS
data for OA.

For the PMF analysis of the PMCAMx OA predictions, we created a matrix **X** in
which each column consists of the hourly PMCAMx-SR predicted concentrations of
POA, SOA-sv and SOA-iv, biomass burning POA, biomass burning SOA, anthropogenic
SOA, biogenic SOA and long range transport (OA transported from outside the model
domain). The material in each bin with $C* \leq 10^2$ μg m$^{-3}$ was included in the PMF
analysis as an independent OA component. The OA in volatility bins with higher
saturation concentrations was excluded, because its particulate phase concentrations are
negligibly small or zero. Table S1 provides a complete list of the 27 OA components
used in our PMF analysis. We implicitly assume that each OA component is "chemically
different" from the others.

## 2.2 Particulate Source Apportionment Technology

PSAT is a computationally-efficient source apportionment algorithm for studying PM
source apportionment contributions (Wagstrom et al., 2008) extended by Skyllakou et al.
(2014) to include OA simulated with the VBS. Skyllakou et al. (2018) used (PSAT)
together with the volatility basis set framework (Donahue et al., 2006) to estimate the age
of the OA components in Europe during the same period as in this study. In this
application, the PSAT algorithm works in parallel with the CTM and provides the "fresh"



secondary components (first generation), the products of two generations of reactions, etc. These results of Skyllakou et al. (2018) are used here.

In order to apply PMF to the results of PSAT we generated a matrix **X** which includes the hourly concentration of OA components categorized as "fresh", long range transport OA, fresh biogenic SOA, fresh anthropogenic SOA, aged (second and later)

SOA-sv and SOA-iv with each saturation concentration ($C^*$) ranging from 0.01 to 100. Table S2 shows the 19 OA components used in this PSAT-based PMF analysis.

**2.3 Positive Matrix Factorization (PMF)**

PMF (Paatero and Taaper, 1994) is a bilinear model that has been used for the

quantification of the sources of airborne particulate matter measurement. PMF decomposes the 'observation' matrix X into two matrices **G** and **F**:

$$x_{ij} = \sum_{k=1}^{p} g_{ik} f_{kj} + e_{ij} \tag{1}$$

where $x_{ij}$ are the measurements used as the PMF inputs, $g_{ik}$ are the contributions of sources, $f_{kj}$ are the factor profiles and $e_{ij}$ the residuals of the analysis. The subscript $i$

corresponds to time, $j$ to the compounds and $p$ is the number of factors. Factor profiles and time series are derived by the PMF model minimizing the objective function $Q$:

$$Q = \sum_{i=1}^{m} \sum_{j=1}^{n} \left( \frac{e_{ij}}{u_{ij}} \right)^2 \tag{2}$$

where $u_{ij}$ are the data uncertainties with the constraint **G** and **F** matrices to be positive. In this study we used 5%, 10% and 20% uncertainty for each data point of matrix **U** and we

did not observe significant differences in the results. For this reason, a 10% uncertainty is assumed for each data point.

In this work, we first created the matrices **X** and **U** in proper format consistent with EPA PMF v5.0. Then, we ran PMF assuming 2, 3, 4 factors and so on. For the selection of the number of factors that best describes our data we used a series of metrics.

We first examined the change of $Q/Q_{exp}$ for each solution. $Q$ is the sum of the squares of the scaled residuals and $Q_{exp}$ represents the ideal value if the residuals were the same as the uncertainty assumed for each data point. We then examined the residuals of the model as a function of the number of factors. We also estimated the correlation



coefficients of the time series of the factors determined by PMF. If a pair of factors was
strongly correlated, we reduced the number of factors. We also checked the composition
of each factor. If there is a pair of factors with similar composition, this solution is
rejected. For the chosen solution, we also investigated the change of factor profile with
positive and negative values of *fpeak*. If the factor profiles are insensitive to the *fpeak*
choice, we proceeded with *fpeak* equal to zero.


### 2.4 The Multilinear Engine (ME-2)

In selected cases, we also used the multilinear engine (ME-2) algorithm (Paatero, 1999)
implemented within the toolkit Sofi (Source Finder) developed by Canonaco et al.
(2013). We used ME-2 in areas in which an HOA factor was not found by PMF. For the
selection of the number of factors, we followed similar steps as with PMF. The main
difference with PMF analysis is that we introduced the vector $F_j$ (factor profile) which
includes only the contribution of POA components with the rest of the OA components
having zero contribution to this factor. The ME-2 algorithm $a$ value determines the extent
to which the output factor profile can vary from the factor profile which we provide
(Canonaco et al., 2013). We used $a$=0.1 for our analysis. We also examined different
values of $a$ ranging from 0 to 0.3, but our results were not sensitive to that choice.

## 3. Results and discussion

### 3.1 PMCAMx-SR results

The predicted average OA at the ground level was 1.8 μg m$^{-3}$ during the simulation
period with average concentrations as high as 4 μg m$^{-3}$ in central and north-eastern
Europe (Fig. S1a). The average concentration of POA was 1.4 μg m$^{-3}$ with the highest
levels predicted in northern Europe (Fig. S1b). SOA levels were higher in central Europe
(Fig. S1c). Details about these predictions can be found in Fountoukis et al. (2011; 2014)
and Theodoritsi and Pandis (2018).

### 3.2 Application of PMF to PMCAMx-SR OA

We first analyse the PMCAMx-SR OA predictions in Melpitz (Germany) because there
were AMS measurements and corresponding PMF results available in this site during the



same period. The average PMCAMx-SR predicted OA in that site was 4.2 μg m$^{-3}$, while

the observed OA was 5.3 μg m$^{-3}$. PMCAMx-SR predicted that long-range transported

OA contributed 24%, biogenic SOA 23%, SOA from SVOCs and IVOCs 20%,

anthropogenic SOA 18%, biomass burning SOA 10%, POA 3% and biomass burning

POA 2% to the total OA.

The 2-factor PMF solution explained the PMCAMx-SR OA predictions. A 2-

factor solution had also been found by Poulain et al. (2014) during their PMF analysis of

the field measurements in the same period. The first PMCAMx-SR factor includes a

variety of secondary OA components: biomass burning SOA (10%), anthropogenic SOA

(20%), biogenic SOA (45%) and SOA-sv and SOA-iv (20%) (Fig. 1). It contains mostly

SOA (around 95%) and therefore will be called "SOA factor" (Fig. 1). The second factor

contains mostly (50%) OA from long range transport and therefore will be called 'LRT

factor'. The remaining 50% of the LRT-factor is mainly anthropogenic SOA (14%),

SOA-sv and SOA-iv (24%) and biomass burning SOA (10%). The SOA-factor

contributed 53% to the predicted OA while the LRT-factor 47%. The concentrations of

both factors were quite variable (Fig. 2), but the SOA factor fluctuated more than the

LRT factor.

            During the same period two factors were identified analyzing the AMS

measurements in Melpitz: low-volatility oxygenated OA (LV-OOA) and a semi-volatile

oxygenated OA (SV-OOA) factor (Poulain et al., 2014). The average diurnal profile of

the PMCAMx-SR SOA factor follows the same pattern as SV-OOA (Fig. 3a) with higher

values during the night. The PMCAMx-SR LRT factor is less than the AMS LV-OOA

factor during the day. These differences can be due to model errors or can be actual

differences in the PMF analysis of the two data sets.

            The above results are quite encouraging. This analysis of the two data sets

suggests that the PMCAMx-SR PMF analysis provides results that are similar with the

corresponding analysis of the AMS measurements. Both approaches result in two

oxygenated OA factors. Even more the AMS LV-OOA factor appears to correspond to

the LRT factor of PMCAMx-SR, and the AMS SV-OOA factor to the PMCAMx-SR

SOA factor. We will return to the Melpitz dataset in a subsequent section focusing on

OOA. In the next two sections we focus on the major primary OA factors.



### 3.2 Biomass burning organic aerosol

In this section, we examine whether the PMCAMx-SR factor which represents biomass burning (bbOA) sources consists of only bbOA components. In St. Petersburg (Russia) PMCAMx-SR predicted hourly bbOA levels exceeded 200 µg m$^{-3}$ due to the nearby fires affecting the site on May 4-5 (Fig. 4a). During the full month in this site, the average contribution of fresh biomass burning OA to the total OA was approximately 65%. During the fire period (4-5 of May) the bbOA contribution was 96%. The 4-factor PMF solution seems to best represent the PMCAMx-SR OA predictions in St. Petersburg. PMF determined a factor which consists of 96% biomass burning POA and low contributions from biogenic SOA and biomass burning SOA components (Fig. 5). This factor will be called "bbPOA factor". In this case, the bbPOA factor includes little else. Comparing the time series of the bbPOA factor and the bbPOA predicted by PMCAMx-SR we estimated a fractional error of 5% and a fractional bias of -3% (Table S3).

In Catania (Italy) the hourly bbPOA concentration exceeded 35 µg m$^{-3}$ during May 15-17 due to nearby fires (Fig. 4b). During the fire period, the contribution of bbPOA to OA reached 94%. During the full month, the average bbPOA contribution to the total OA was 40%. A 3-factor PMF solution was selected in this case. PMF determined a factor with 93% biomass burning POA and the remaining 7% was biomass burning SOA (4%), biogenic SOA (2%) and anthropogenic SOA (1%) in (Fig. 5). The corresponding normalized error when the time series of the bbOA factor was compared to the PMCAMx-SR bbOA predictions was 11% in this case.

In Majden (FYROM) fires contributed up to 15 µg m$^{-3}$ of bbPOA on May 25-26 and bbPOA was 75% of the OA during the fire period (Fig. 4c). The average bbPOA contribution to OA was 14% during the simulation period. The 3-factor PMF solution best fit our data. PMF identified a factor consisting of 81% bbPOA, 11% biogenic SOA, 4% long range transport OA, 2% biomass burning SOA and 2% anthropogenic SOA (Fig. 5). The corresponding normalized error comparing this factor against the actual bbPOA was 24% due to the mixing of the fresh bbPOA with secondary OA from other sources by the PMF.

In Cabauw bbPOA contributed 8% to OA according to PMCAMx-SR with an average concentration of 0.4 µg m$^{-3}$. There were no major fires nearby and the predicted





hourly bbPOA concentration was always less than 3 μg m$^{-3}$. The bbPOA in this case was included by the PMCAMx-SR PMF in a "bbPOA/SOA" factor. This factor is called bbPOA/SOA because it consisted of bbPOA and SOA components. The PMF analysis did not give a bbPOA factor even when 5 factors were used. The same lack of a bbPOA factor was found in the analysis of the PMCAMx-SR OA in Melpitz and Finokalia. The maximum predicted hourly concentration of bbPOA in Melpitz was 0.5 μg m$^{-3}$ and in Finokalia 0.1 μg m$^{-3}$. The bbPOA in these areas was less than 2% of the OA.

In areas affected by major fires (St. Petersburg, Catania and Majden) the maximum predicted hourly concentration of bbSOA was 12, 6.5 and 5.7 μg m$^{-3}$, respectively. In all areas examined in this study bbSOA was included mainly in one of the OOA factors which will be discussed in detail in the next section. This is due to the fact that the temporal evolution of bbSOA is closer to that of the other SOA components. Therefore, the contribution of biomass burning determined by PMF represents a lower estimate of the impact of fires to OA in a receptor since it includes only a small fraction of the bbSOA.

### 3.3 Primary organic aerosol

The ability of PMF to identify the fresh POA from sources other than biomass burning is explored in this section. POA according to PMCAMx-SR contributed 10% to OA during May in St. Petersburg. The 4-factor PMF solution included a factor which consisted of 67% POA (Fig. 6). The remaining was SOA-sv and SOA-iv (9%), biogenic SOA (6%), anthropogenic SOA (5%), biomass burning POA (8%) and biomass burning SOA (5%). We call this "POA factor", but it clearly includes other OA components. For the purposes of our analysis, we consider that PMF identifies a POA factor if there is a factor containing more than 60% POA. The POA factor and PMCAMx-SR POA concentrations were well correlated to each other (R$^2$=0.99, Fig. S2). The average concentration of the POA factor was 1.1 μg m$^{-3}$ and of the actual POA 0.9 μg m$^{-3}$. The normalized error of the POA factor compared to the PMCAMx-SR POA was 34% (Table S4).

The highest contribution of POA to total OA was predicted in Majkow Duzy in central Poland and it was 50%. In this location, the POA contributed 90% to the corresponding POA factor (Fig. 6). The remaining was biogenic SOA (3%), long range



transport OA (4%), anthropogenic SOA (1%), biogenic SOA (1%) and biomass burning SOA (1%). The average concentration of the POA factor was 3.2 μg m⁻³, while the

PMCAMx-SR actual POA was 3.4 μg m⁻³. The normalized error of the POA factor 10% in this case (Table S4).

In rural and remote sites (Cabauw, Melpitz and Finokalia) POA contributed around 3% to the total OA according to PMCAMx-SR. In Cabauw the 3-factor solution included factors which contained 6%, 11% and 10% POA, respectively. In the 4-factor

solution POA contributed 12%, 10%, 5% and 0% to the factors. In these areas, PMF did not separate the POA from the rest of the OA components.

**3.4 PMF error for primary OA components**

The above analysis of the bbOA and POA factors suggests that the corresponding PMF

error does depend on the magnitude of the contribution of the corresponding source to the total OA levels. Higher relative errors are estimated when a source contributes less to the total OA. To better quantify the corresponding dependence of the error on the magnitude of the source we used the PMF solutions in a number of locations and we also artificially scaled up and down the predicted bbOA and POA in certain locations (St.

Petersburg, Maiden, Catania, Cabauw, and Majkow Duzy) and repeated the PMF analysis. The results are summarized in Fig. 7.

The normalized mean error of the bbPOA estimated by the PMF is less than 30% when the bbPOA contributes more than 20% to the total OA in the area. The error is reduced to less than 20% for contributions higher than 30%. On the other hand, when the

bbPOA represents 10-20% of the total OA the PMF error can be up to 50%. When biomass burning contributes less than 10% the error increases to a factor of 2-3. Please note that in these cases, the absolute error is still reasonable and the PMF correctly predicts that bbOA is a relatively small component of OA.

The uncertainty in POA from other sources appears to be a little higher than that

of bbPOA probably because PMF mixes it with other sources that have similar temporal profiles. When the POA represents more than 20% of the OA, the PMF error is less than 40%. The errors can be up to a factor of 2, when the POA is less than 20% of the OA.



### 3.5 Oxygenated Organic Aerosol

In this section we try to determine the characteristics that differentiate the two OOA factors that are often present in ambient AMS data analysis. One hypothesis is that the two OOA factors contain different OA components (e.g. anthropogenic versus biogenic). A second hypothesis is that one represents the semivolatile and the other the low-volatility OA components. The third hypothesis is that these two factors have different

degrees of aging (one is relatively fresh SOA and other SOA that has undergone multiple generations of oxidation).

The two PMCAMx-SR OOA factors in all areas consist mainly of multiple SOA components. The first OOA factor determined by PMF analysis of PMCAMx-SR OA predictions contains mainly OA from long range transport. This factor was determined in

all areas examined.

In St. Petersburg long range transport (LRT) OA was 11% of the OA according to PMCAMx-SR. The 4-factor solution included a factor which contained 55% LRT-OA and is described here as the "LRT factor" (Fig. 8). In Majden the contribution of LRT-OA to total OA was 25%. In this area PMF determined a LRT factor with 68% long

range transported OA. In Catania LRT OA contributed 29% to OA and the LRT factor consists of 70% LRT-OA. In Cabauw and Melpitz the contribution of long range transport OA was 21% and 24% and the corresponding LRT factors consist of 58% and 48% LRT-OA, respectively. During May, the highest contribution of LRT-OA to total OA was determined in Finokalia and it was around 40%. In this site, the LRT-OA

contributed 87% to the LRT factor (Fig. 8). Thus, the contribution of highly aged OA transported from outside the domain to the LRT factor ranges from approximately 50% to 90% for the areas examined.

The second OOA factor determined in all areas contains SOA-sv and SOA-iv, anthropogenic SOA, biomass burning SOA and biogenic SOA (Fig. 9). We call this

"SOA factor" because it mostly includes SOA produced inside the modeling domain. In Catania, PMF combines bbSOA (20% contribution to SOA factor), aSOA (20%) and SOA-sv and SOA-iv (30%) in the SOA factor because the time series of these OA components follow a similar pattern during the simulation period (Fig. S3). This is also the case in the other areas (Majden, Melpitz and Finokalia, Figs. S4-S6) examined. The



contribution of each SOA component to the SOA factor depends on the examined area. Therefore, the SOA factor consists of a mixture of contributions from various anthropogenic and biogenic sources.

        While the two OOA factors both include a mixture of all SOA components (Figures 8 and 9) the LRT factor is dominated by the highly aged OA transported to

Europe from outside the domain, while the SOA factor includes mainly SOA produced over Europe. Therefore, the hypothesis that PMF separates the SOA components based on their sources (e.g. biogenic versus anthropogenic) is not supported by our results.

**3.5.1 Volatility of OOA factors**

We analyzed the volatility distribution of the two PMCAMx-SR OOA factors predicted by PMCAMx-SR in order to examine whether these factors include OA components with different volatility. In Melpitz the volatility distribution of the SOA factor peaks at effective saturation concentration equal to 1 µg m$^{-3}$ (Fig. 10a). 90% of the OA in this factor has effective saturation concentration ($C^*$) higher or equal to 1 µg m$^{-3}$. On the

other hand, the LRT factor is dominated by components with $C^*$ equal to 0.01 and 0.1 µg m$^{-3}$, contributing 80% to the factor. In Finokalia the highest mass fraction of the LRT factor has effective saturation concentration equal to 0.01 µg m$^{-3}$ (Fig. 10c). The LRT factor in this case contains almost exclusively low volatility OA. The SOA factor includes both low volatility and semivolatile components. In St. Petersburg, Catania and

Majden the results for the volatility distribution of LRT and SOA factor were between those in St. Petersburg and in Finokalia (Fig. S7).

        These results suggest that both factors have components covering a wide range of volatilities and their volatility distributions overlap. However, the LRT factor has on average lower volatility than the SOA factor. These suggest that the PMF does not

separate these factors exclusively based on the volatility of the corresponding components. For example, in Melpitz both factors include a lot of OA with $C^*$ equal to 1 µg m$^{-3}$.




### 3.5.2 The degree of aging of OOA factors

We applied PMF analysis to PSAT results, separating all the SOA components into two subcategories: first generation and later generation products (second, third, etc.), to investigate whether the degree of chemical processing differentiates the two OOA

factors.

In Melpitz the first PMCAMx/PSAT factor consists of 63% first generation OA and 37% later generation OA and is called the "less aged factor" (Fig. 11). The second factor includes 97% later generation OA and can be described as the "more aged factor".

In the more remote site of Finokalia, we determined two factors which both

contain aged OA. We characterize the first factor as "extremely aged" because highly aged long range transport OA dominated this factor (98%) (Fig. 11). The second factor is characterized as "very aged" containing 32% later generation aSOA, 54% later generation SOA from semi-volatile and intermediate volatility organic compounds and only 14% first generation SOA. These results are consistent with the analysis of

Hildebrandt et al. (2010) who argued that the OA behaviour in Finokalia is quite different that in continental European sites and that the two OOA factors are quite similar to each other. This is also predicted by PMCAMx-SR suggesting that the model is consistent with that interpretation of the measurements.

### 3.5.3 Comparison of OOA factors of PMF and ME-2 analysis

In this section, we compare the two OOA factors determined by PMF and ME-2 analysis in order to estimate the change of these factors when ME-2 analysis is used. In ME-2 we used the "correct" POA factor (forced the model to assume 100% contribution of POA to the POA factor). Moving from PMF to ME-2, the changes of the composition of the SOA

and LRT factor were minor in all examined areas. Figures S8 and S9 illustrate the two OOA factors in Melpitz and in Finokalia when PMF and ME-2 are used. Thus, the above conclusions for the two OOA factors do not change when ME-2 is used. The gain of the use of ME-2 analysis is that a better separation of primary sources is obtained if of course the correct POA fingerprint is used.




## 4. Conclusions

We analyzed for the first time, to the best of our knowledge, the organic aerosol composition predictions of a chemical transport model (PMCAMx-SR) using positive matrix factorization in an effort to better understand the results of PMF analysis of ambient organic aerosol AMS measurements. The high-level results of our analysis are quite consistent with those of the corresponding field studies; we find similar number and characteristics of factors for a number of sites in Europe. This consistency indicates that the analysis of the model results can be used as a first order interpretation of the various factors often reported in field data PMF analysis. These factors include the hydrocarbon like OA and biomass burning OA and two oxygenated organic OA factors. Cooking OA was not included as a source in the emissions inventory used, so it cannot be studied at this stage.

The primary OA factor (which corresponds to the hydrocarbon-like OA in AMS analysis) of the PMCAMx-SR predictions usually contains not only primary OA compounds but also secondary components or biomass burning OA. These additional components represent on average one third of the factor mass. The average error of using HOA instead of POA is around 25% in the cases examined and increases when the POA contribution to OA decreases. PMF identifies a POA factor in the PMCAMx-SR predictions when this group of sources contributes more than 10% to the OA and is one of the top three sources.

PMF determines a biomass burning OA factor in all areas influenced by major nearby fires. In these cases of major fire influence, the biomass burning primary OA factor consists of around 90% biomass burning primary OA. The error in the bbPOA factor is less than 30%, when biomass burning contributes more than 20% to the average OA. The biomass burning secondary OA is grouped always with secondary OA components and only a small fraction of it is included in the biomass burning factor. Therefore, the bbOA factor provides a lower limit of the impact of fires on the OA of an area.

Our analysis suggests that PMF has difficulties identifying sources that contribute approximately 10% or less to the total OA during the period of the analysis. The use of ME-2 constraining the primary OA factor (which contains 100% contribution from



primary OA) provides a better separation of primary and secondary sources reducing the contribution of POA to the two oxygenated OA factors. However, this assumes perfect knowledge of the "fingerprint" of the POA factor.

The two oxygenated OA factors both contain a series of SOA components from both anthropogenic and biogenic sources. The first oxygenated OA factor includes mainly highly aged OA transported from outside Europe, but also highly aged secondary OA from sources in Europe that has undergone multiple generations of oxidation. The second oxygenated OA factor contains SOA from volatile, semi-volatile, and intermediate volatility anthropogenic and biogenic organic compounds. The exact contribution of these OA components to each OA factor depends on the site. In rural continental areas (like Melpitz) the first oxygenated OA factor includes highly aged secondary OA and the second mostly "fresh" first-generation secondary organic compounds. On the other hand, in remote sites such as in Finokalia in Crete, both oxygenated OA factors include organic components that have undergone two or more generations of aging. This suggests that the PMF determines the two extremes of the chemical processing of the OA present in the site during the measurements and reports them as the corresponding OOA factors.

The two oxygenated OA factors have most of the time overlapping volatility distributions and therefore their characterization as low and high volatility that has been used in the literature is probably misleading. This is consistent with the measurements of Paciga et al. (2016) in Paris and Louvaris et al. (2017) in Athens. However, the more aged factor has lower average volatility than the fresh secondary OA factor.

**Acknowledgements**

This work has been supported by the US Environmental Protection Agency (EPA) Center for Air, Climate and Energy Solutions (CACES) (grant number R835873). The authors would like to thank K. Florou for her assistance with PMF.

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






**Figure 1.** SOA and LRT factor profiles resulting from the PMF analysis of the
PMCAMx-SR OA predictions in Melpitz.




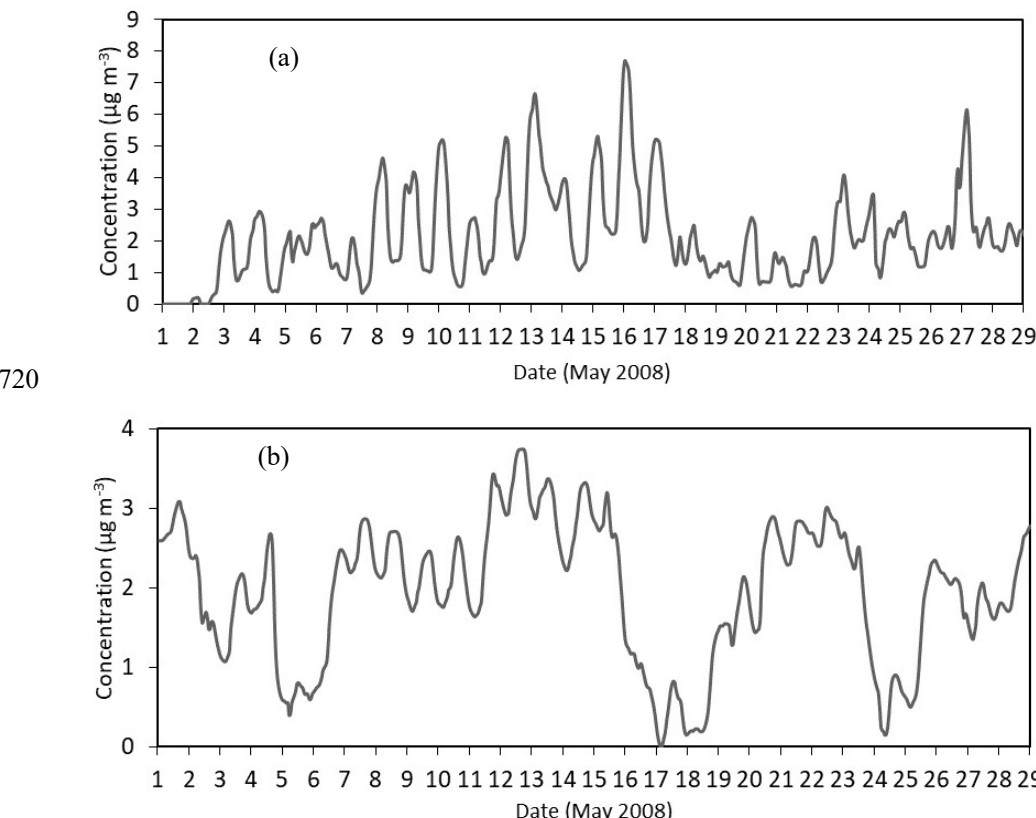

**Figure 2.** PMCAMx-SR factor time series of the: (a) SOA and (b) LRT factors in
Melpitz during May 2008.




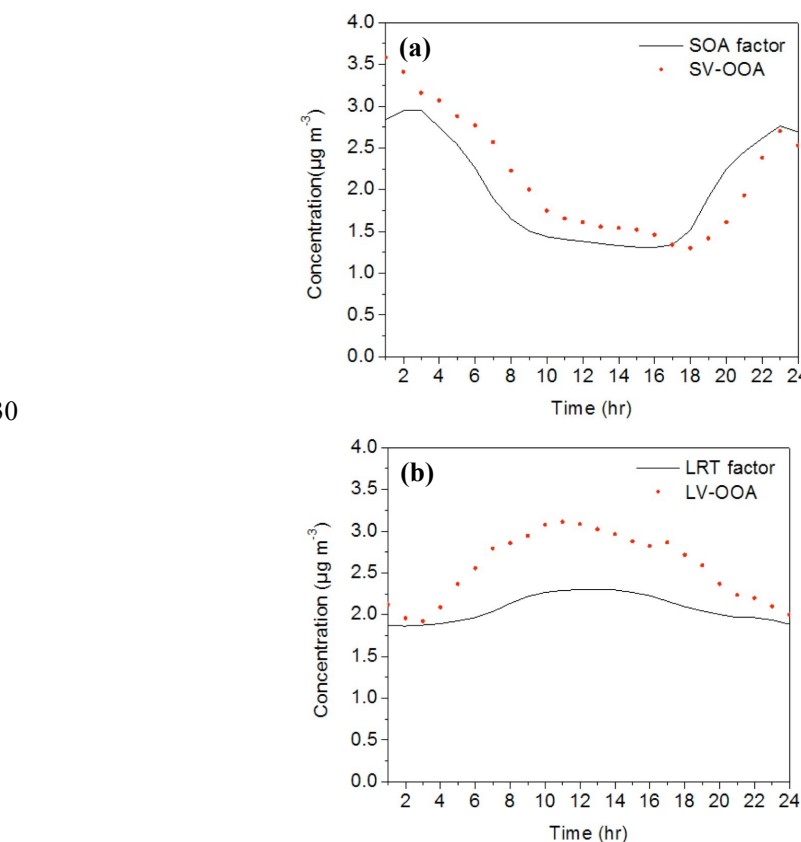

**Figure 3.** Comparison of average diurnal profiles of factors of PMF analysis of
PMCAMx-SR results and PMF analysis of AMS measurements in Melpitz: (a) SOA
factor and SV-OOA and (b) LRT factor and LV-OOA.




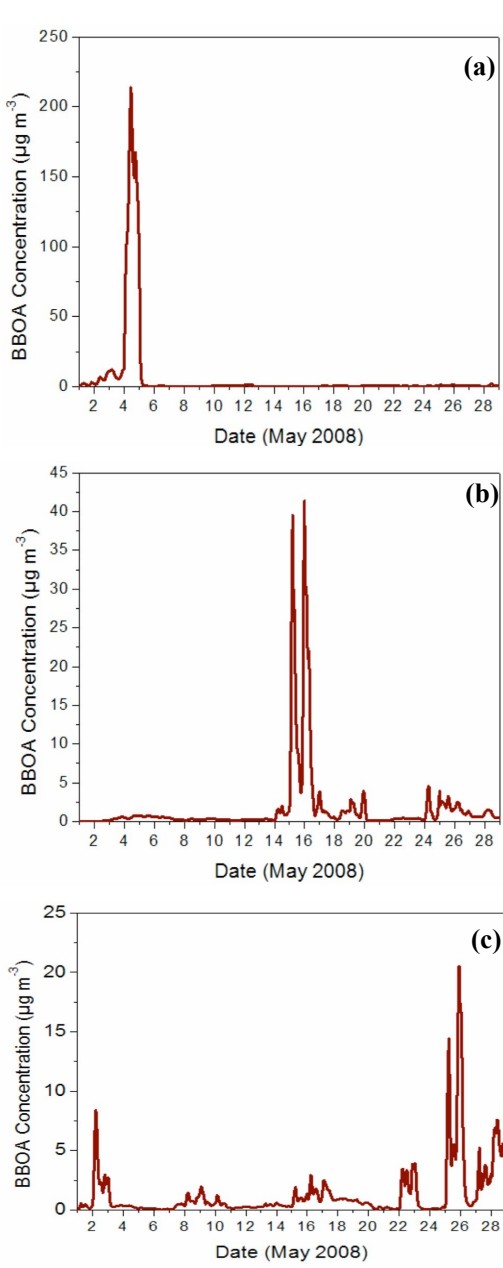

**Figure 4.** Time series of predicted biomass burning OA by PMCAMx-SR during May
2008: (a) St. Petersburg (Russia), (b) Catania (Italy) and (c) Majden (FYROM).





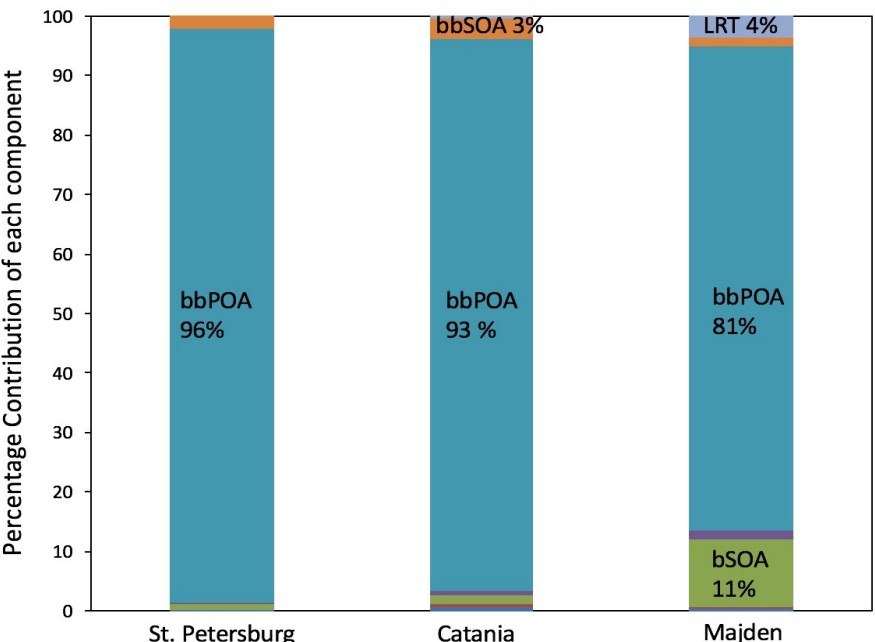


**Figure 5.** Contribution of each OA component to the PMCAMx-SR bbPOA factor in St. Petersburg, Catania. and Majden during May 2008.






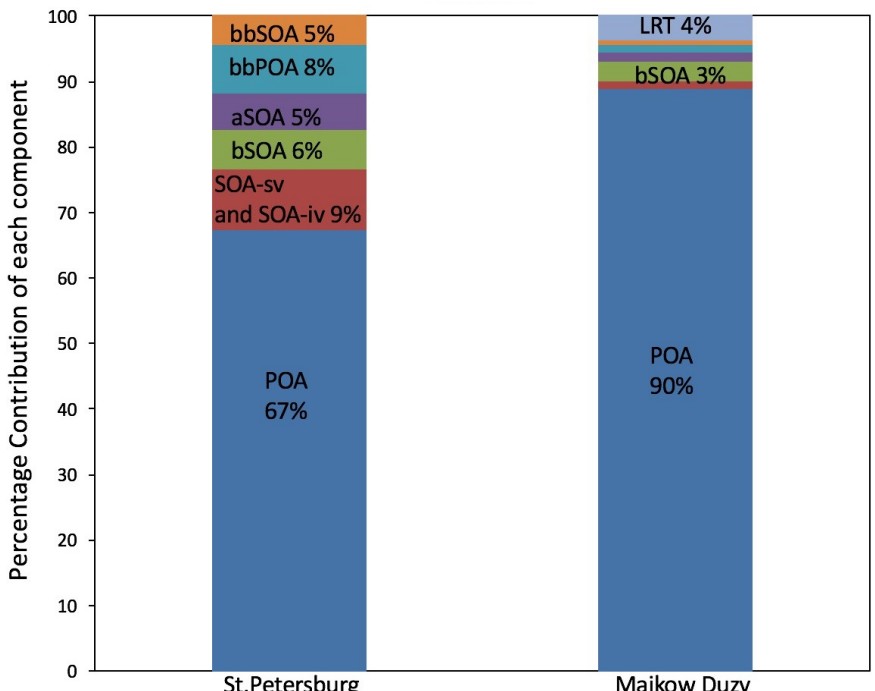

**Figure 6.** Contribution of each OA component to the PMCAMx-SR PMF POA factor in

St. Petersburg (Russia) and Majkow Duzy (Poland) during May 2008.






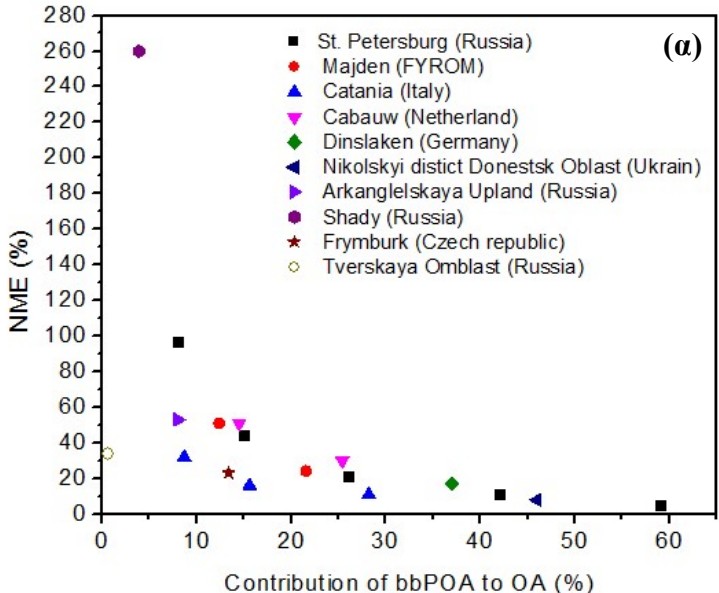

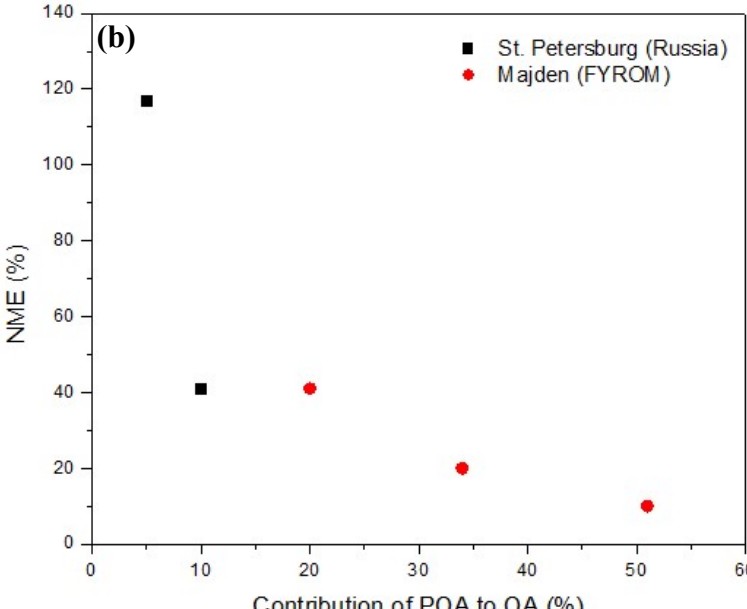


**Figure 7.** PMF normalized error (%) for (α) bbPOA and (β) POA for various locations as a function of their contribution to OA.




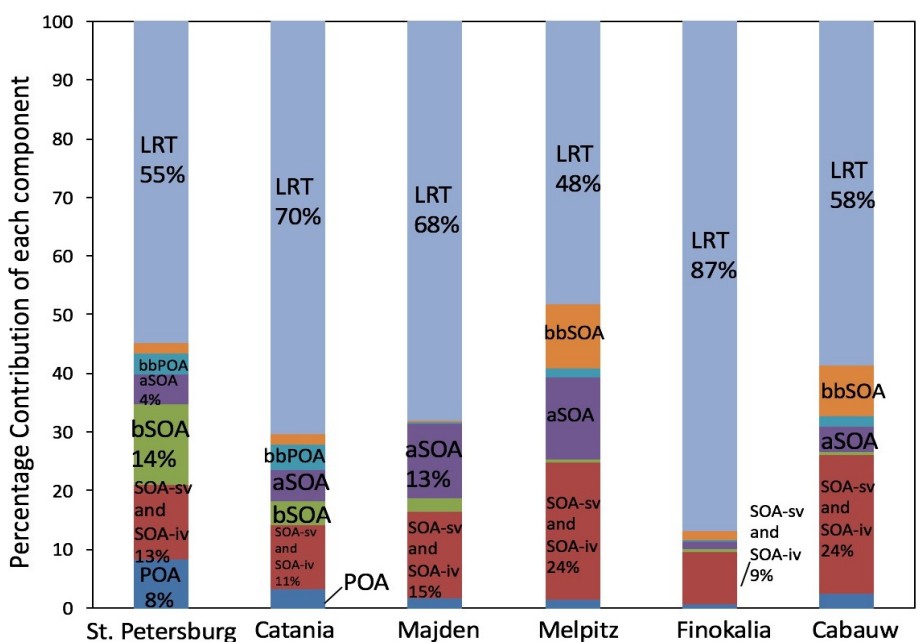

**Figure 8.** Contribution of each OA component to PMCAMx-SR LRT factor in St.

Petersburg, Catania, Majden, Melpitz and Finokalia during May 2008.



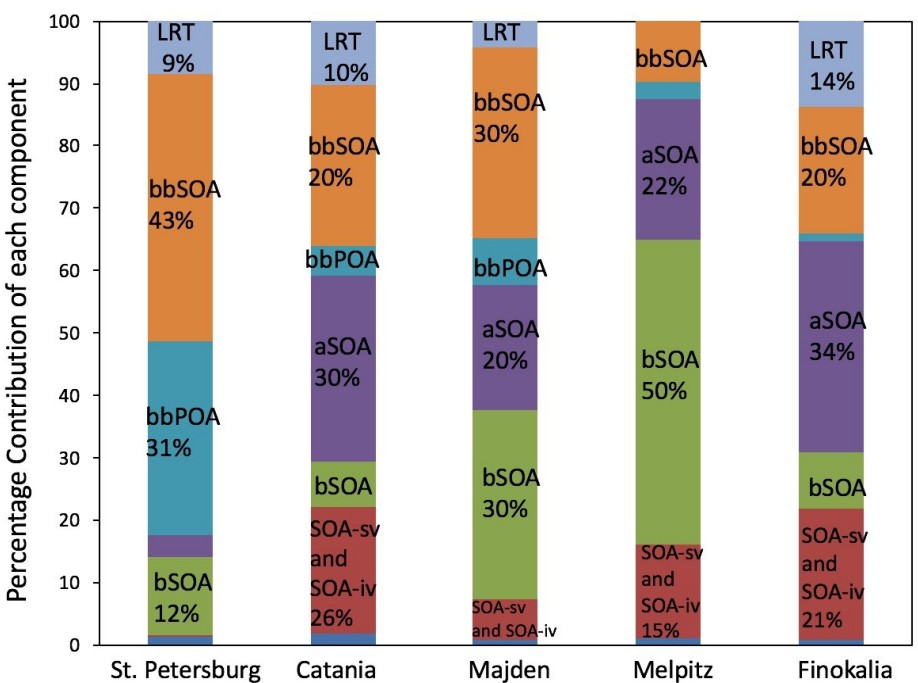

**Figure 9.** Contribution of each OA component to the PMCAMx-SR SOA factor in St. Petersburg, Catania, Majden, Melpitz and Finokalia during May 2008.



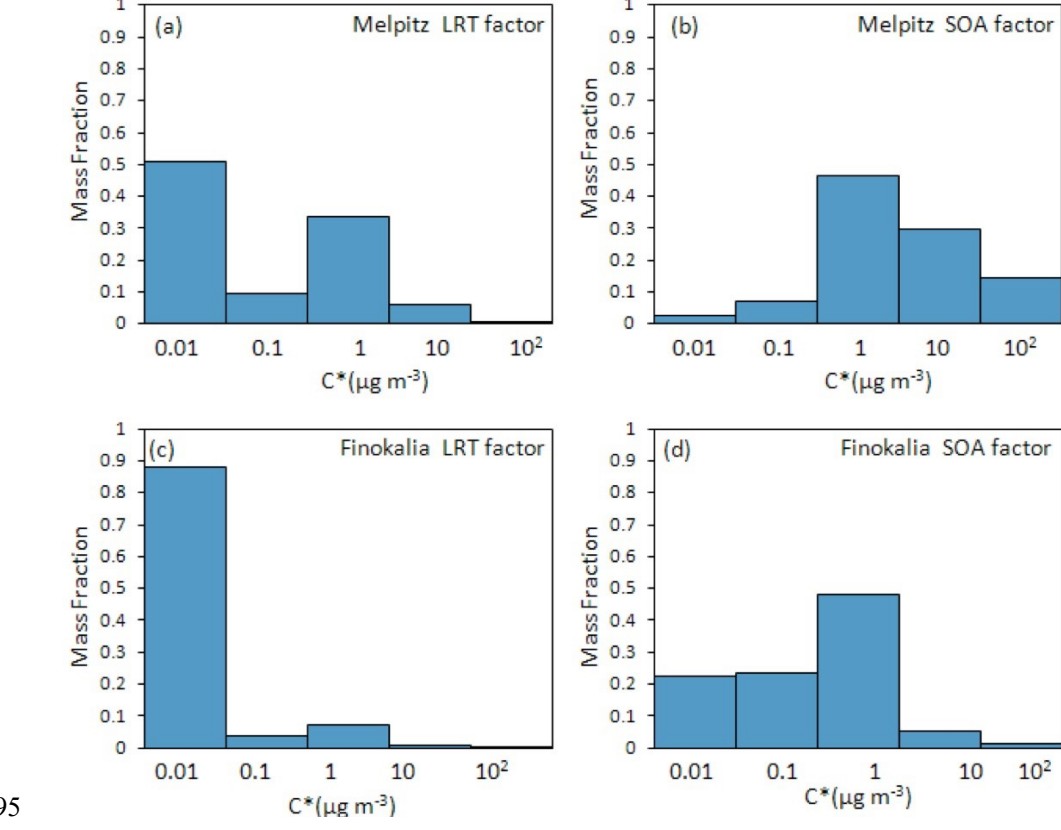


**Figure 10.** Volatility distribution of the: (a) LRT factor in Melpitz, (b) SOA factor in Melpitz, (c) LRT factor in Finokalia, and (d) SOA factor in Finokalia.






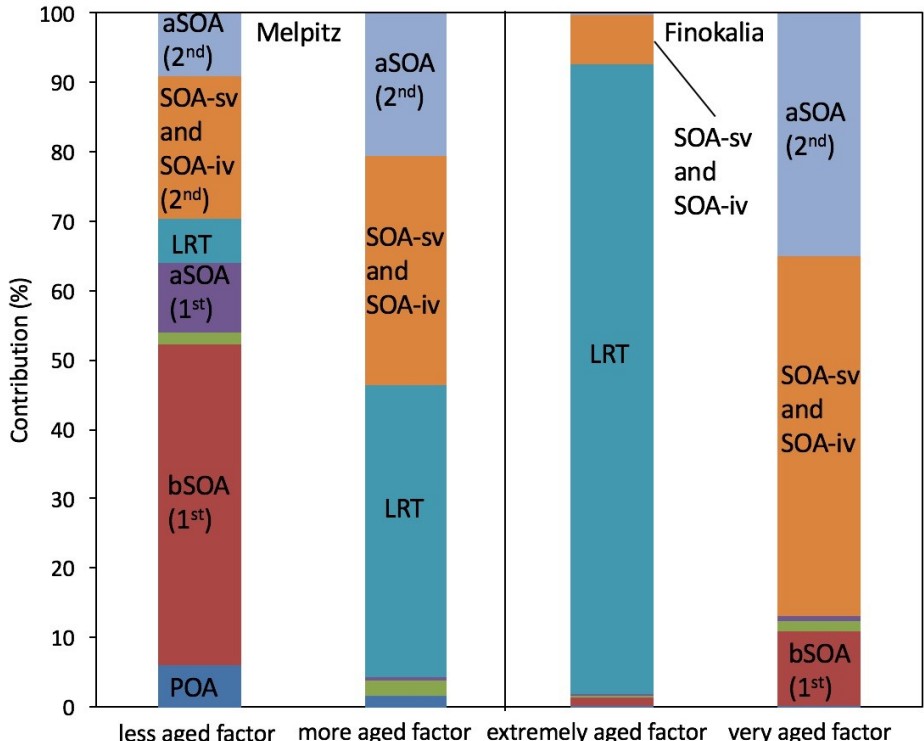

**Figure 11.** Contribution of first generation and second plus later generations of SOA components to each factor in Melpitz and Finokalia during May 2008.
