# Peer review of "Positive Matrix Factorization of Organic Aerosol: Insights from a Chemical Transport Model"

_Atmospheric Chemistry and Physics, 2018_

## Referee Comment (RC1) · Anonymous Referee #1 · 6 Sep 2018

Drosatou et al. present a study in which chemical transport model predictions of organic aerosol over Europe are used in a PMF (and ME-2) analysis to determine what types of organic aerosol factors exist in model predictions. The use of a chemical transport model means that the PMF factors can be directly examined in terms of their sources and identity. The major findings include the composition of POA factors in terms of potential SOA contributions and expected error as well as identification of 2 types of SOA or OOA-like factors in a variety of locations. They show that the two types of SOA do not separate anthropogenic and biogenic SOA and the separation is mainly based on age. This is a useful analysis and comments below focus on two main areas.

[Figure]

Main comments:

1. Methodology

a. How would results be different if source information (for example the chemical identity of POA and bbPOA) was not used in the PMF analysis of model output? If the volatility of the model-predicted OA was the only chemical information in the PMF analysis, would you get similar results? This may provide insight into how results from this work translate to AMS analysis in which sources may not be very chemically distinct due to fragmentation.

b. PMF appears to have been performed on a site by site basis. Can this be clarified? The analysis generally always leads to two SOA (or OOA) factors, but the composition of the SOA factors varies by site. How many SOA factors would be obtained if all locations from the model were used in one PMF analysis? An analysis along these lines could help inform questions in the second main comment regarding how different the two OOA factors are in different locations or studies.

c. What information is introduced by the PMF analysis of model output that is not otherwise available? Could the same "factors" be obtained by determining how much POA, SOA-sv, bSOA, aSOA, etc correlate or covary and making two groups?

d. How was the boundary condition OA at the edge of the domain specified and evaluated? If the boundary OA was not assigned a C* of 0.01 ug/m3, would it have ended up in a different factor?

2. Meaning of two SOA or OOA factors

a. Is the proliferation of terms in literature (LO/MO-OOA, OOA-1/2, LV/SV-OOA) indicating true site to site variability in the OOA components or is it just a nomenclature choice?

b. The authors argue that the designation of the two AMS factors based on volatility is somewhat misleading due to overlap in their volatility. I was not convinced that this

designation was misleading (based on Figure 10) but do agree that it is a simplification. What is the best description of the two factors given that they likely overlap on many metrics (volatility, O:C, age, etc) and age, O:C, and volatility covary?

c. How should CTMs evaluate their predictions compared to AMS data beyond SOA vs OOA? Can the analysis here be used to provide a range of agreement where models can be assumed "in agreement?"

Minor comments:

1. Lines 22-28 of the abstract are useful, but could be condensed. Mentioning the fraction of the POA factor that is secondary (e.g. lines 477-478) would be even more useful.

2. Lines 81-104 are missing the MO- and LO-OOA designations (Xu et al. 2015 PNAS https://doi.org/10.1073/pnas.1417609112) in the discussion and how those fit with the other AMS PMF factors from literature.

3. Line 134: what version of SAPRC was used?

4. Paragraph starting at line 250: Clarify that there was no observed POA or bbPOA factor in observations or model for Melpitz.

5. Figure 3: Could boundary OA and POA+S/IVOC-SOA be added to panel (b)? Could SOA (excluding boundary and S/IVOC-SOA) be added to panel (a)? How much value does the PMF factor bring compared to classifying boundary and S/IVOC-SOA as one type and all other SOA as one type?

6. Figure 7: How were the locations chosen?

7. Figure 7: What would you expect the NME to be for typical urban, suburban, or rural conditions (add to plot)?

8. Did the model include any aqueous SOA? Where would that appear in the PMF analysis?

[Figure]

Editorial comments:

1. Line 126: "a regional three-dimensional CTM" is duplicated within the sentence.

2. Figure 1: Needs (a) and (b) labels or titles.

3. Figure 2: Could be on same panel in different colors

4. Figure 4: Could go in SI

5. Figure 11: switch columns 3 and 4 so that both Melpitz and Finokalia data reads as increasing age going left to right.

---

## Referee Comment (RC2) · Anonymous Referee #2 · 14 Sep 2018

This paper summarizes PMF analysis conducted on OA simulated from a chemical transport model and compared with AMS PMF factors for a single summer month. The authors find similar number and categorization of PMF factors as AMS, and provides further insight for the modeled factors. First, the primary (HOA-like) factor often contains some SOA and biomass OA. Second, two OOA components found are likely more oxygenated and less oxygenated, but are not always separated in volatility space. Finally (among other conclusions), SOA from various sources can be lumped into a single PMF factor. The manuscript is well-written and easy-to-read. Many of the technical decisions (e.g., regarding the PMF error matrix for simulations) seem well-justified, and there are additional insights regarding errors in source apportionment

(mixing of sources or not extracting the source altogether) during periods of minor contributions. There results are of great interest to the atmospheric chemistry community and is therefore recommended for publication with minor revisions.

General comments:

The authors do not seem to explicitly mention this, but PMF appears to have been conducted separately at each site. One reason for applying site-specific PMF to measurements is that anthropogenic or biogenic SOA can be different at each site, but in these simulations they are effectively the same (e.g., a lumped species with C* belonging to anthropongenic POA is chemically the same across sites). However, the site-specific PMF is still desirable here for capturing site-specific proportion of source classes in each factor, and for comparison with site-specific AMS PMF.

The assumption that the components are "chemically different" is mentioned a couple of times, but it is not further discussed. For instance, the lumped species differ in C* or reactivity with OH (depending on source class) so in many cases they are already treated as being chemically different. For the remaining lumped species, it's not unlikely that species from different source classes grouped in the same bin C* are likely structurally different. However, a real chemical difference with respect to gross properties should not necessarily be required by PMF either, as it is possible to use isotope-labeled compounds in the analysis (which source-tagging effectively does here).

Is the simulated OA size fraction used for PMF analysis equivalent to PM2.5 or what might be more directly comparable to the submicron fraction measured by AMS?

Can the authors remark on the fact that the model, which does not include aqueous-phase or condensed-phase chemistry, generates factors which agree on a "high-level" with AMS factors? Is it that condensed-phase processes do not provide sufficient differentiation from covariations fixed by vapor-phase processes? Or could it a limitation of the sites/period studied?

The authors have previously published work integrating 2D-VBS into PMCAMx in the European domain; the model used here is apparently different and referred to as PMCAMx-SR - but the citation refers to a manuscript in preparation so it is hard to understand some of the details. How is this model different and why was the 1-D VBS selected? For this comparison with AMS, comparison of O/C ratios would have been useful to show further correspondence (or differences) between simulated and measured PMF factors (e.g., Aiken et al. 2008, Canagarathna et al. 2015); the authors may wish to add justification for the decision they made here.

Aiken, A. C., Decarlo, P. F., Kroll, J. H., Worsnop, D. R., Huffman, J. A., Docherty, K. S., Ulbrich, I. M., Mohr, C., Kimmel, J. R., Sueper, D., Sun, Y., Zhang, Q., Trimborn, A., Northway, M., Ziemann, P. J., Canagaratna, M. R., Onasch, T. B., Alfarra, M. R., Prevot, A. S. H., Dommen, J., Duplissy, J., Metzger, A.; Baltensperger, U. & Jimenez, J. L. O/C and OM/OC ratios of primary, secondary, and ambient organic aerosols with high-resolution time-of-flight aerosol mass spectrometry, Environmental Science & Technology, 42, 4478-4485, https://doi.org/10.1021/es703009q, 2008.

Canagaratna, M. R., Jimenez, J. L., Kroll, J. H., Chen, Q., Kessler, S. H., Massoli, P., Hildebrandt Ruiz, L., Fortner, E., Williams, L. R., Wilson, K. R., Surratt, J. D., Donahue, N. M., Jayne, J. T., and Worsnop, D. R.: Elemental ratio measurements of organic compounds using aerosol mass spectrometry: characterization, improved calibration, and implications, Atmos. Chem. Phys., 15, 253-272, https://doi.org/10.5194/acp-15-253-2015, 2015.

Others have found instances where the low/high volatility designation of PMF OOA do apply (e.g., Cappa and Jimenez 2010), so the conclusion (line. 512) that the nomenclature is misleading seems to broad - it may be rephrased that statistical separation of OOA by volatility cannot always be assumed?

Cappa, C. D. and Jimenez, J. L.: Quantitative estimates of the volatility of ambient organic aerosol, Atmos. Chem. Phys., 10, 5409-5424, https://doi.org/10.5194/acp-10-

5409-2010, 2010.

Minor comments:

- Should the acronym PMCAMx be defined? For instance, CMAQ (Community Multi-scale Air Quality Modeling System) is typically spelled out when first introduced in a paper.

- Regarding terminology, line 141: "primary organic compounds are all considered to be semi-volatile with $C^*$ ranging from $10^{-2}$ to $10^6$ microg/m$^3$" whereas Donahue, Robinson, and Pandis (2009) define SVOCs to have $C^*$ ranging between $10^0$ and $10^2$ microg/m$^3$.

Donahue, N. M., Robinson, A. L. & Pandis, S. N. Atmospheric organic particulate matter: From smoke to secondary organic aerosol, Atmospheric Environment, 43, 94-106 https://doi.org/10.1016/j.atmosenv.2008.09.055, 2009.

- Figure 7: Is this not more a reflection of the deviation in source apportionment for both measurements and simulation when the source contribution becomes small, rather than error that can be purely attributed to the apportionment from the simulation side (as reflected by normalization to observed values)? Section 3.4 should correspondingly be renamed since "PMF error" can imply many things (error matrix, residual matrix, etc.).

---

## Author Comment (AC1) · 19 Nov 2018

**(1)** *Drosatou et al. present a study in which chemical transport model predictions of organic aerosol over Europe are used in a PMF (and ME-2) analysis to determine what types of organic aerosol factors exist in model predictions. The use of a chemical transport model means that the PMF factors can be directly examined in terms of their sources and identity. The major findings include the composition of POA factors in terms of potential SOA contributions and expected error as well as identification of 2 types of SOA or OOA-like factors in a variety of locations. They show that the two types of SOA do not separate anthropogenic and biogenic SOA and the separation is mainly*

*based on age. This is a useful analysis and comments below focus on two main areas.*

We appreciate the comments and suggestions of the reviewer. Our responses and corresponding changes in the manuscript (in regular font) can be found below after each comment (in italics).

*Main comments:*

**(2)** *How would results be different if source information (for example the chemical identity of POA and bbPOA) was not used in the PMF analysis of model output? If the volatility of the model-predicted OA was the only chemical information in the PMF analysis, would you get similar results? This may provide insight into how results from this work translate to AMS analysis in which sources may not be very chemically distinct due to fragmentation.*

This is an interesting suggestion allowing the PMF to focus just on the volatility of the OA. We have repeated the PMF analysis using only the volatility distributions. We first tried two factors. The corresponding PMF factors included material from all volatility bins. So PMF did not separate the OA into semi-volatile and low volatility material. In the next step we assumed three factors but still the factors included surrogate compounds with a mixture of volatilities. We have added a new sub-section in the Results to discuss the results of this test simulation that further supports our conclusion that PMF does not separate OA components solely based on their volatility.

**(3)** *PMF appears to have been performed on a site by site basis. Can this be clarified? The analysis generally always leads to two SOA (or OOA) factors, but the composition of the SOA factors varies by site. How many SOA factors would be obtained if all locations from the model were used in one PMF analysis? An analysis along these lines could help inform questions in the second main comment regarding how different the two OOA factors are in different locations or studies.*

[Figure]

Indeed, the analysis presented in the original paper was performed for each site separately similarly to the analysis of field campaign measurements. This is now clarified in the manuscript. We have followed the reviewer's suggestion and performed an additional test in which we combined results from all seven sites (all the examined areas in the paper). The application of PMF to this comprehensive set resulted in four factors: fresh biomass burning, other primary OA and two secondary OA factors (fresh and aged SOA). The number and character of the factors were similar with the site by site analysis, but there were differences in the composition and contribution of the factors. The results of this test are now discussed in the revised paper.

**(4)** *What information is introduced by the PMF analysis of model output that is not otherwise available? Could the same "factors" be obtained by determining how much POA, SOA-sv, bSOA, aSOA, etc correlate or covary and making two groups?*

The reviewer is correct; factor analysis methods are in general based on the temporal correlation among the concentrations of different pollutants. However, in their effort to limit the dimensionality of the chemical (or AMS m/z) space, these approaches distribute the pollutants into factors in ways that are by no means transparent. Our goal in this work has been to shed a little more light on what PMF does when it is applied to the AMS organic aerosol data. We have added this brief discussion in the paper.

**(5)** *How was the boundary condition OA at the edge of the domain specified and evaluated? If the boundary OA was not assigned a C\* of 0.01 $\mu$g/m$^3$, would it have ended up in a different factor?*

In general boundary conditions of regional chemical transport models are obtained from the output of similar global models or from some averages of measurements. For the purposes of this work we assumed low volatility constant OA boundary conditions. This choice facilitates the separation of highly aged OA coming from outside the modeling domain from the fresher material that is emitted and/or produced inside the

domain. Obviously, the absolute OA concentrations especially near the boundaries of the domain can be dominated by these boundary conditions. To avoid such issues, we have used sites that are far from the boundaries. Overall, our conclusions are quite robust to the choice of the OA boundary condition values. The effect of our choices of OA boundary conditions is now discussed in the paper.

**(6)** *Meaning of two SOA or OOA factors. Is the proliferation of terms in literature (LO/MO-OOA, OOA-1/2, LV/SV-OOA) indicating true site to site variability in the OOA components or is it just a nomenclature choice?*

We believe that the evolution of the terms used to describe these two factors reflects our understanding (or lack there-of) of the nature of these factors and not so much site to site variability. The use of OOA1 and OOA2 reflected the complete lack of understanding. Then the use of Less and More Volatile OOA showed the beginning of some understanding, but it has probably led to some confusion and a few misconceptions. The next step (use of Less and More Oxidized OA) is probably more accurate. Our work here supports the hypothesis that these factors correspond to Less and More Aged OOA present in each site. We have added this discussion to the corresponding section of the paper.

**(7)** *The authors argue that the designation of the two AMS factors based on volatility is somewhat misleading due to overlap in their volatility. I was not convinced that this designation was misleading (based on Figure 10) but do agree that it is a simplification. What is the best description of the two factors given that they likely overlap on many metrics (volatility, O:C, age, etc) and age, O:C, and volatility covary?*

The use of the volatility-based terminology suggested to most of us that there is a volatility threshold and OA components that are more volatile than this are grouped by PMF in one factor (e.g., SV-OOA) and the less volatile compounds in the second (LV-OOA). Our results both from this theoretical analysis but also from direct volatility

measurements of AMS factors (Paciga et al., 2016; Louvaris et al., 2017) suggest that this is not the case. The so-called semivolatile factor may include very low volatility OA and vice versa the so-called low-volatility factor may include semivolatile material. We believe that the use of more and less oxygenated is safer and that the use of more and less aged will be probably proven to be more accurate.

**(8)** *How should CTMs evaluate their predictions compared to AMS data beyond SOA vs OOA? Can the analysis here be used to provide a range of agreement where models can be assumed "in agreement?"*

This is an excellent question. Our results suggest that the comparison of CTM predictions of POA and fresh biomass burning OA to the corresponding AMS results is meaningful if these are major sources and taking into account the uncertainties estimated here. The comparison of the less and more volatile OA predicted by CTMs to the corresponding OOA factors is probably not a good idea. Summation of the two OOA factors into just OOA appears to be quite safe based on our results here. On the other hand, if a CTM can keep track of the age of OA the comparison of more and less aged predicted OA to the two OOA factors could be potentially useful. We have added this discussion about model evaluation to the paper.

*Minor comments:*

**(9)** *Lines 22-28 of the abstract are useful, but could be condensed. Mentioning the fraction of the POA factor that is secondary (e.g. lines 477-478) would be even more useful.*

We have followed the reviewer's suggestion and rewrote this part of the abstract.

**(10)** *Lines 81-104 are missing the MO- and LO-OOA designations (Xu et al. 2015 PNAS https://doi.org/10.1073/pnas.1417609112) in the discussion and how those fit with the other AMS PMF factors from literature.*

We have tried to keep the terminology used in the corresponding studies to both show the evolution of the nomenclature but also the rather confusing picture. We have added a sentence indicating the correspondence of the terms (for example OOA-1, LV-OOA and MO-OOA have been used for the same factor in most studies).

**(11)** *Line 134: what version of SAPRC was used?*

A version of SAPRC99 extended to include the VBS species was used in this work. This information has been added to the text.

**(12)** *Paragraph starting at line 250: Clarify that there was no observed POA or bbPOA factor in observations or model for Melpitz.*

We have added this clarification in the revised paper.

**(13)** *Figure 3: Could boundary OA and POA+S/IVOC-SOA be added to panel (b)? Could SOA (excluding boundary and S/IVOC-SOA) be added to panel (a)? How much value does the PMF factor bring compared to classifying boundary and S/IVOC-SOA as one type and all other SOA as one type?*

The major point of this figure and the corresponding example of the PMF analysis in Melpitz is that the analysis of the ambient AMS dataset and that of the PMCAMx VBS predictions results in the same number of factors with a very similar diurnally-averaged behavior. This supports our hypothesis that the PMF analysis of the PMCAMx predictions can help us understand better the results of the PMF/AMS analysis. One can try different combinations of the predictions of PMCAMx and compare them to the results of PMF/AMS (see for example Fountoukis et al., 2014 for such an effort), but this is outside the scope of this work. Please note that we are not suggesting that PMF should be applied to CTM results. CTMs (as we show in this work too) provide directly information about sources of OA without the need of PMF. A brief discussion of this point has been added.

[Figure]

**(14)** *Figure 7: How were the locations chosen?*

We have added a description of our criteria for the choice of these locations. Briefly, Majkow Duzy (Poland) has the highest predicted contribution of POA to OA. St. Petersburg, Catania and Majden are three locations in different environments with high bbOA levels during the simulation period. Melpitz, Cabauw and Finokalia were chosen because there are AMS measurements available for the simulation period and they also cover quite different environments. The other sites in Figure 7 were chosen because they had different predicted bbOA/levels and they could help us get information for the full range of values.

**(15)** *Figure 7: What would you expect the NME to be for typical urban, suburban, or rural conditions (add to plot)?*

Based on our analysis the actual contribution of bbOA to the total OA is more important than the type of environment for Europe at least. If a site (even if it is urban) is influenced by a major nearby fire contributing significantly to the OA then PMF does well in quantifying its impact. If on the other hand if a rural site is only marginally affected by far away fires then the corresponding normalized error can be significant.

**(16)** *Did the model include any aqueous SOA? Where would that appear in the PMF analysis?*

No, this version of PMCAM did not include aqueous SOA production. The treatment by PMF of such OA that has been produced by a different pathway is an interesting question for future work.

*Editorial comments*

**(17)** *Line 126: "a regional three-dimensional CTM" is duplicated within the sentence.*

We have rewritten this sentence.

**(18)** *Figure 1: Needs (a) and (b) labels or titles.*

We have added the corresponding labels.

**(19)** *Figure 2: Could be on same panel in different colors.*

We would prefer to keep them separate. The figure becomes rather confusing when these two timeseries are in the same panel.

**(20)** *Figure 4: Could go in SI.*

We have followed the reviewer's suggestion and moved this figure to the SI.

**(21)** *Figure 11: switch columns 3 and 4 so that both Melpitz and Finokalia data reads as increasing age going left to right.*

We have switched the order of the two Finokalia factors for consistency with the Melpitz factors.

---

## Author Comment (AC2) · 19 Nov 2018

**(1)** *This paper summarizes PMF analysis conducted on OA simulated from a chemical transport model and compared with AMS PMF factors for a single summer month. The authors find similar number and categorization of PMF factors as AMS, and provides further insight for the modeled factors. First, the primary (HOA-like) factor often contains some SOA and biomass OA. Second, two OOA components found are likely more oxygenated and less oxygenated, but are not always separated in volatility space. Finally (among other conclusions), SOA from various sources can be lumped into a single PMF factor. The manuscript is well-written and easy-to-read. Many of the technical*

[Figure]

*decisions (e.g., regarding the PMF error matrix for simulations) seem well justified, and there are additional insights regarding errors in source apportionment (mixing of sources or not extracting the source altogether) during periods of minor contributions. These results are of great interest to the atmospheric chemistry community and is therefore recommended for publication with minor revisions.*

We appreciate the positive assessment of our work. Our responses and corresponding changes in the manuscript (in regular font) can be found below after each comment (in italics).

*General comments:*

**(2)** *The authors do not seem to explicitly mention this, but PMF appears to have been conducted separately at each site. One reason for applying site-specific PMF to measurements is that anthropogenic or biogenic SOA can be different at each site, but in these simulations they are effectively the same (e.g., a lumped species with C\* belonging to anthropogenic POA is chemically the same across sites). However, the site-specific PMF is still desirable here for capturing site-specific proportion of source classes in each factor, and for comparison with site-specific AMS PMF?*

This is a good point also made by the first reviewer (Comment 3). Indeed, the PMF analysis presented in the original paper was performed for each site separately similarly to the standard analysis of field campaign measurements. This is now clarified in the manuscript. We have complemented this site-by-site analysis with analysis of the combined data in all sites. The application of PMF to this comprehensive set resulted in four factors: fresh biomass burning, other primary OA and two secondary OA factors (fresh and aged SOA). These could explain well the overall dataset. The number and character of the factors were similar with the site by site analysis, but there were differences in the composition and contribution of the factors. The results of this test are now discussed in a new section in the revised paper.

[Figure]

**(3)** *The assumption that the components are "chemically different" is mentioned a couple of times, but it is not further discussed. For instance, the lumped species differ in C\* or reactivity with OH (depending on source class) so in many cases they are already treated as being chemically different. For the remaining lumped species, it's not unlikely that species from different source classes grouped in the same bin C\* are likely structurally different. However, a real chemical difference with respect to gross properties should not necessarily be required by PMF either, as it is possible to use isotope-labeled compounds in the analysis (which source-tagging effectively does here).*

This is an important aspect of our analysis that needs further clarification because it is the most important difference of our CTM-based approach and the AMS/PMF analysis of field data. As we provide PMF with the concentrations of 27 different predicted OA surrogate components, we implicitly assume that the corresponding measurement technique or techniques can separate and quantify these components. For the AMS, this may not be the case as two OA components (e.g., processed bbOA and aged SOA) may have quite similar AMS spectra. Of course, other measurement techniques, like the one mentioned by the reviewer, have different capabilities. We now provide a little more discussion about this assumption underlying our work.

**(4)** *Is the simulated OA size fraction used for PMF analysis equivalent to PM2.5 or what might be more directly comparable to the submicron fraction measured by AMS?*

We have used PM1 for our analysis for consistency with the AMS measurements. However, the difference in predicted OA in the PM2.5 and PM1 range is small in PMCAMx so our conclusions are also valid for PM2.5. This point is now explained in the revised paper.

**(5)** *Can the authors remark on the fact that the model, which does not include aqueous phase or condensed-phase chemistry, generates factors which agree on a "high-level"*

*with AMS factors? Is it that condensed-phase processes do not provide sufficient differentiation from covariations fixed by vapor-phase processes? Or could it a limitation of the sites/period studied?*

This is a very interesting question also posed by the first reviewer (Comment 16). One could speculate that it may grouped by PMF together with the other aged OA. Unfortunately, we cannot test this hypothesis with the results of the current version of the model that does not include aqueous-phase production of SOA. It is clearly a good topic for future work.

**(6)** *The authors have previously published work integrating 2D-VBS into PMCAMx in the European domain; the model used here is apparently different and referred to as PMCAMx-SR - but the citation refers to a manuscript in preparation so it is hard to understand some of the details. How is this model different and why was the 1-D VBS selected? For this comparison with AMS, comparison of O/C ratios would have been useful to show further correspondence (or differences) between simulated and measured PMF factors (e.g., Aiken et al. 2008, Canagarathna et al. 2015); the authors may wish to add justification for the decision they made here.*

The reviewer is correct, the version of the model used here (PMCAMx-SR) is based on the 1D-VBS, similarly to the regular PMCAMx. Its major difference from its sister model is its ability to simulate separately the primary and secondary OA from different sources. Therefore, one can use different volatility distributions and aging schemes for organic compounds from different sources. This allows us in this work to use more up-to-date information about the bbOA properties. Use of the OA from the 2D-VBS in a similar exercise is in an excellent idea and could allow one to include the O:C in the analysis. This is the topic of ongoing work.

**(7)** *Others have found instances where the low/high volatility designation of PMF OOA do apply (e.g., Cappa and Jimenez 2010), so the conclusion (line. 512) that the nomen-*

*clature is misleading seems to broad - it may be rephrased that statistical separation of OOA by volatility cannot always be assumed?*

The use of the volatility-based terminology suggests that there is a volatility threshold and OA components that are more volatile than this are grouped by PMF in one factor (e.g., SV-OOA) and the less volatile compounds in the second (LV-OOA). Our results both from this theoretical analysis but also from direct volatility measurements of AMS factors (Paciga et al., 2016; Louvaris et al., 2017) suggest that this is not the case. The so-called semivolatile factor may include very low volatility OA and vice versa the so-called low-volatility factor may include semivolatile material. We have rephrased the statement to indicate that it may be misleading in at least some cases.

*Minor comments*

**(8)** *Should the acronym PMCAMx be defined? For instance, CMAQ (Community Multiscale Air Quality Modeling System) is typically spelled out when first introduced in a paper.*

We have added the definition of the acronym PMCAMx (Particulate Matter Comprehensive Air Quality Model with extensions).

**(9)** *Regarding terminology, line 141: "primary organic compounds are all considered to be semi-volatile with C\* ranging from $10^{-2}$ to $10^6$ $\mu g/m^3$ whereas Donahue, Robinson, and Pandis (2009) define SVOCs to have C\* ranging between $10^0$ and $10^2$ $\mu g/m^3$.*

We have rephrased this sentence that may confuse some readers about the definition of the term "semi-volatile".

**(10)** *Figure 7: Is this not more a reflection of the deviation in source apportionment for both measurements and simulation when the source contribution becomes small, rather than error that can be purely attributed to the apportionment from the simu-*

*lation side (as reflected by normalization to observed values)? Section 3.4 should correspondingly be renamed since "PMF error" can imply many things (error matrix, residual matrix, etc.).*

The reviewer is right in general, however in our case the "measurement" error is zero as we use predicted values as inputs to the PMF algorithm. So this error is all due to the source apportionment algorithm. Our analysis suggests that this can be quite significant (a factor or 2 or more) for the smaller OA sources, so the corresponding estimates should be used with caution. We have changed the title of Section 3.4 to "PMF source apportionment error".